



# Effects of point source emission heights in WRF–STILT: a step towards exploiting nocturnal observations in models

Fabian Maier[1,2], Christoph Gerbig[3], Ingeborg Levin[1], Ingrid Super[4], Julia Marshall[5] and Samuel Hammer[1,2]

[1]Institut für Umweltphysik, Heidelberg University, INF 229, 69120 Heidelberg, Germany
[2]ICOS Central Radiocarbon Laboratory, Heidelberg University, Berliner Straße 53, 69120 Heidelberg, Germany
[3]Department Biogeochemical Systems, Max Planck Institute for Biogeochemistry, Hans-Knöll-Straße 10, 07745 Jena, Germany
[4]Department of Climate, Air and Sustainability, TNO, P.O. Box 80015, 3508 TA Utrecht, the Netherlands
[5]Deutsches Zentrum für Luft- und Raumfahrt (DLR), Institut für Physik der Atmosphäre, Oberpfaffenhofen, Germany

*Correspondence to*: Fabian Maier (Fabian.Maier@iup.uni-heidelberg.de)



**Abstract.** An appropriate representation of point source emissions in atmospheric transport models is very challenging. In the Stochastic Time Inverted Lagrangian Transport model (STILT), all point source emissions are typically released from the surface, meaning that the actual emission stack height plus subsequent plume rise is not considered. This can lead to erroneous

predictions of trace gas concentrations, especially during nighttime when vertical atmospheric mixing is minimal. In this study we use two WRF–STILT model approaches to simulate fossil fuel $CO_2$ (ff$CO_2$) concentrations: (1) the standard "surface source influence (SSI)" approach, and (2) an alternative "volume source influence (VSI)" approach, where nearby point sources release $CO_2$ according to their effective emission height profiles. The comparison with [14]C-based measured ff$CO_2$ data from two-week integrated afternoon and nighttime samples collected at Heidelberg, 30 m above ground level, shows that the root-

mean-square deviation (RMSD) between modelled and measured ff$CO_2$ is indeed almost twice as high during night (RMSD = 6.3 ppm) compared to the afternoon (RMSD = 3.7 ppm) when using the standard SSI approach. In contrast, the VSI approach leads to a much better performance at nighttime (RMSD = 3.4 ppm), which is similar to its performance during afternoon (RMSD = 3.7 ppm). Representing nearby point source emissions with the VSI approach could, thus, be a first step towards exploiting nocturnal observations in STILT. To further investigate the differences between these two approaches, we

conducted a model experiment in which we simulated the ff$CO_2$ contributions from 12 artificial power plants with typical annual emissions of one million tons of $CO_2$ and with distances between 5 and 200 km from the Heidelberg observation site. We find that such a power plant must be more than 50 km away from the observation site in order for the mean modelled ff$CO_2$ concentration difference between the SSI and VSI approach to fall below 0.1 ppm.





## 1 Introduction

The Integrated Carbon Observation System (ICOS) research infrastructure was established to set up a dense European monitoring network of high-precision greenhouse gas measurements of concentrations and fluxes, therewith providing the observational basis to better understand the European carbon budget (Heiskanen et al., 2021). In Europe, one major challenge is the quantification of anthropogenic fossil fuel $CO_2$ (ff$CO_2$) emissions, but similarly important is to understand "their redistribution among the atmosphere, ocean and terrestrial biosphere in a changing climate" (Friedlingstein et al., 2020). If the

share of ff$CO_2$ in the total continental signal is modelled correctly, the remaining biogenic share can be used as a top-down constraint on the continental biospheric $CO_2$ fluxes (Basu et al., 2016). In this study, we use the term ff$CO_2$ to refer not only to $CO_2$ emissions resulting from the combustion of fossil fuels but also to fossil $CO_2$ emissions which occur during cement production. A well-established approach to determine the regional ff$CO_2$ component in the observed atmospheric $CO_2$ concentration is via $\Delta^{14}CO_2$ measurements (e.g., Levin et al., 2003). Since $CO_2$ emissions from fossil fuel combustion are

devoid of $^{14}C$ (the half-life of $^{14}C$ is 5700 years (Currie, 2004)) the atmospheric $\Delta^{14}CO_2$ depletion measured in polluted areas relative to clean background air allows the regional (or "recently added") ff$CO_2$ surplus to be determined. Many studies have used this approach at various urban and rural sites (e.g., Levin et al., 2008; Turnbull et al., 2015; Wenger et al., 2019). Two-week integrated air samples as well as hourly flask samples are collected at ICOS class-1 stations for $^{14}C$ analysis to estimate regional ff$CO_2$ concentrations (Levin et al., 2020), thus helping to separate biospheric from fossil $CO_2$ fluxes e.g. in an inverse

modelling framework (Wang et al., 2018; Basu et al., 2020).

Estimating ff$CO_2$ fluxes from atmospheric $CO_2$ and $^{14}C$ measurements within an inverse modelling framework requires a correct representation of the atmospheric transport and mixing processes. Geels et al. (2007) evaluated five different Eulerian atmospheric transport models with continuous $CO_2$ observations from various European sites, as well as aircraft flask samples,

and showed that the model predictions are much better in the afternoon hours during well-mixed atmospheric conditions than during stable nocturnal conditions. That is why they recommend to only use afternoon observations from low altitude sites to constrain $CO_2$ sources or sinks. Also, Lagrangian transport models like the Stochastic Time-Inverted Lagrangian Transport model (STILT) are very sensitive to the representation of the planetary boundary layer height (PBLH). STILT determines the sensitivity of atmospheric trace gas mixing ratios at an observation site to upwind surface fluxes (Lin et al., 2003). This so-

called footprint defines the catchment area of the observation site and is, by default, sensitive to emissions from the bottom half of the planetary boundary layer (PBL). In STILT it is assumed that surface emissions are instantaneously mixed by turbulence in the bottom half of the PBL within one model time-step. Gerbig et al. (2008) compared radiosonde-derived mixing heights with mixing heights derived from the European Center for Medium-Range Weather Forecasts (ECMWF) meteorological data for two European summer months in 2005 and used STILT to assess the propagated uncertainty in the

$CO_2$ mole fraction. During daytime, they found no significant relative bias between radiosonde and ECMWF-derived mixing heights, but a relative standard deviation of about 40 %. However, nighttime situations showed a relative bias of more than 50



% with a relative standard deviation of almost 100 %. The authors showed that already the 40 % uncertainty in daytime mixing heights resulted in $CO_2$ mole fraction uncertainties of on average 3 ppm during the two summer months studied, which corresponds to about 30 % of the simulated biogenic signals.


In STILT there is an additional problem, namely the incorrect representation of point source emissions. Point source emissions are often released from chimneys whose stack height can be above the bottom half of the PBL during night, depending on the meteorological situation. However, in STILT the default is that all emissions, including point sources, are released from the ground and mixed into the bottom half of the PBL. Under stable conditions this can result in large overestimations of

concentrations near the surface and large underestimations of concentrations above the PBL.

In Central Europe, about 45 % of the $ffCO_2$ emissions are released from point sources (Super et al., 2020), underlining the potential impact of these elevated emissions on downwind measurement sites. Figure 1 shows the distributions of $ffCO_2$ point sources in Europe and illustrates how close some of the ICOS stations are located to these big $ffCO_2$ point source emitters. An

attempt was made to avoid station locations with strong emissions in the vicinity when designing the ICOS atmosphere station network. Nevertheless, there are eight ICOS class-1 or class-2 stations for which the emissions of the energy and industrial $ffCO_2$ point sources within a 50 km x 50 km box around the station sum up to more than one million tons of $CO_2$ per year. This calls for an appropriate representation of point source emissions when modelling $ffCO_2$ concentrations at these ICOS stations.


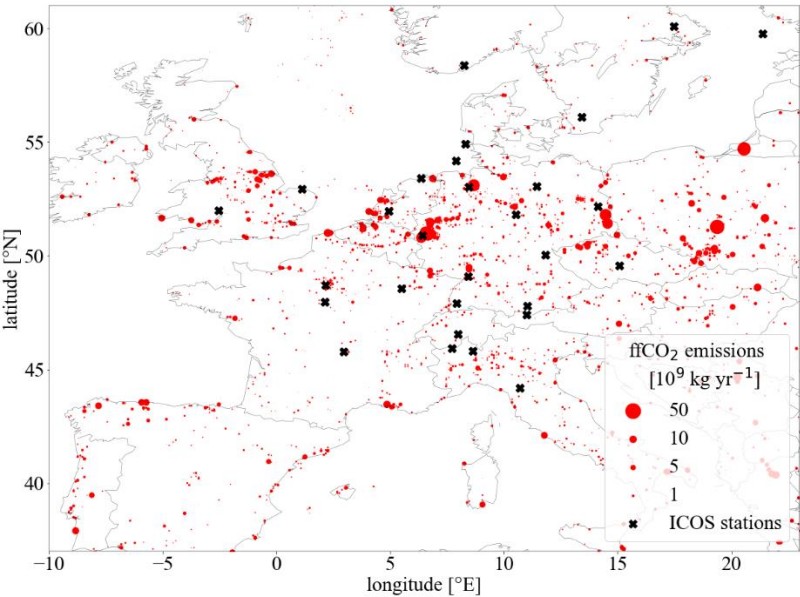

**Figure 1: European $ffCO_2$ point source emissions according to Super et al. (2020, red dots) and the locations of ICOS atmosphere class-1 and class-2 stations (black crosses).**





Together, the inadequate representation of atmospheric transport processes during stable (nighttime) conditions and the
incorrect release of point source emissions at ground level restrict the use of observational data in STILT inversions to daytime
situations only. Atmospheric transport processes are more reliably modelled for daytime situations and the exact representation
of the point source emission heights is less important when atmospheric mixing is strong (Brunner et al., 2019). However,
using nighttime observations would have several advantages: *(1) More data*: Usually (e.g., at ICOS stations) continuous
greenhouse gas measurements are available at all hours of the day and night. A restriction to the afternoon hours means that
about 75 % of the available observations are not used. *(2) Different source mixtures*: Nighttime (morning and evening)
measurements sample different source mixtures than afternoon measurements. As an example, diffuse sources such as heating
or traffic are more dominant during nighttime and the morning or evening rush hours, respectively. *(3) Diurnal cycles*:
Including nighttime observations could help to constrain diurnal emission patterns. For instance, Super et al. (2021) showed
that a correct representation of temporal emission profiles is essential for inverse modelling in urban areas. An important goal
for the future should therefore be to also exploit nighttime observations in modelling frameworks.

In this study, we investigate the effect of a more realistic representation of point source emission heights. Instead of using the
classical approach in STILT, where footprints describe the surface influence on the bottom half of the PBL (hereafter called
"surface source influence" approach), we introduce the so-called "volume source influence" approach that allows point source
emissions to be better represented in STILT. In the volume source influence (VSI) approach, point source emissions are
distributed to pre-defined height intervals in the catchment area of the observation site. If the height profile of a point source
emission is known, its contribution at the observation site can then be estimated with this VSI approach. In the following, we
first evaluate the VSI approach against the standard surface source influence (SSI) approach (Sect. 3.1). For this, we model
the ffCO$_2$ concentrations for our study site, Heidelberg, from July 2018 to June 2020 by applying (a) the SSI approach and (b)
the VSI approach to the point source emissions in the surroundings of Heidelberg. We then compare modelled ffCO$_2$
concentrations to ffCO$_2$ estimates based on two-week integrated daytime and nighttime $\Delta^{14}CO_2$ data from samples collected
in Heidelberg during these two years. In a second step, we investigate how the surface and volume source influence approaches
behave for point sources at increasing distances from the observation site during different atmospheric conditions (Sect. 3.2).
For this, we placed 12 artificial ("pseudo") power plants at distances of 5 to 200 km from our study site and modelled their
mean contribution during different atmospheric conditions.

## 2 Methods

### 2.1 Site description

Heidelberg is a medium-sized city with about 160,000 inhabitants located in the Upper Rhine Valley in south-western
Germany. It is part of the Rhine-Neckar metropolitan area with the heavily industrialized cities Mannheim (310,000
inhabitants) and Ludwigshafen (170,000 inhabitants) about 15–20 km northwest of Heidelberg. The measurement site is in the



northern outskirts of Heidelberg at the Institute of Environmental Physics, which is located on the university campus. There, continuous greenhouse gas measurements and $^{14}CO_2$ sampling are performed with the sample air intake on the roof of the Institute's building about 30 m above the ground. A more detailed description of the Heidelberg measurement site can be found in Hammer (2008). Figure 2 shows the main $ffCO_2$ point sources in the surroundings of Heidelberg. The largest nearby $ffCO_2$

emitters are the coal-fired power plant in Mannheim, the BASF company in Ludwigshafen, a cement production facility (Heidelberg Zement) south of Heidelberg, and a combined heat and power station about 500 m north of the measurement site.

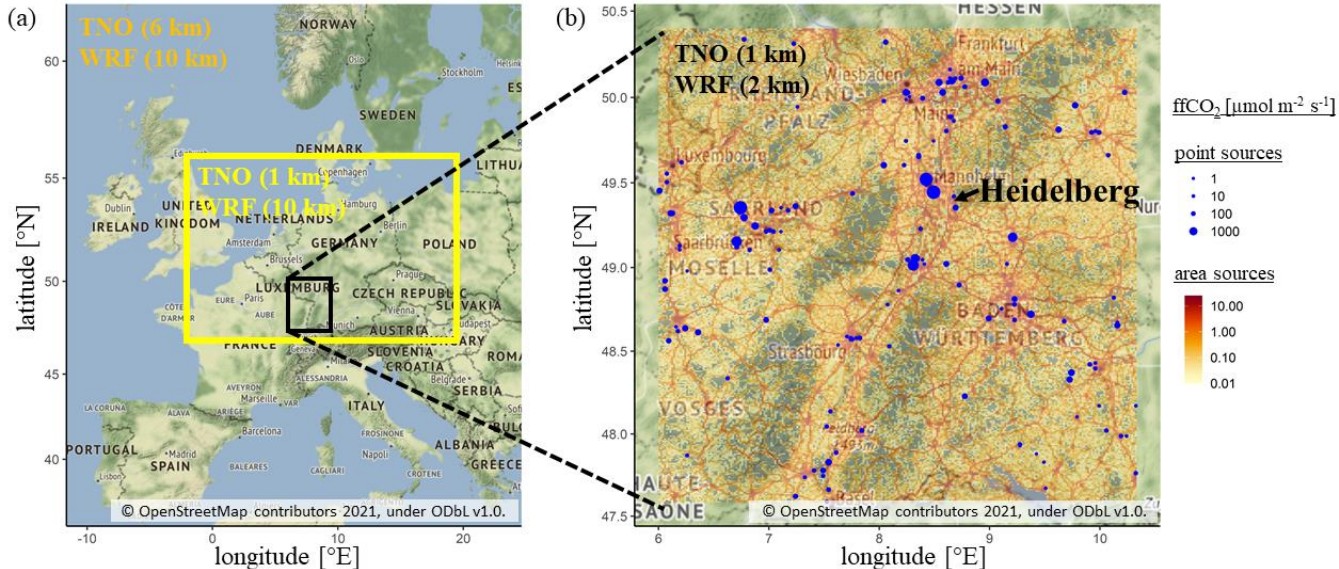

**Figure 2: (a) Model domain and spatial resolution (in brackets) of nested WRF meteorological fields and TNO emission inventories.**
**The right panel (b) shows a zoom into the Rhine Valley with TNO area (orange) and point (in blue) source emissions (from Super et al., 2020). Source: Map tiles by Stamen Design, under CC BY 3.0 (http://maps.stamen.com/terrain/). Data © OpenStreetMap contributors 2021. Distributed under the Open Data Commons Open Database License (ODbL) v1.0.**

## 2.2 Model configuration

We use the coupled Weather Research and Forecasting–Stochastic Time-Inverted Lagrangian Transport model WRF–STILT to simulate hourly $ffCO_2$ concentrations for our measurement site in Heidelberg. STILT is a well-established particle dispersion model, which uses the mean advection scheme from the Hybrid Single-Particle Lagrangian Integrated Trajectory (HYSPLIT) model (Stein et al., 2015), but with a different representation of turbulence. A detailed description of the WRF–STILT model can be found in Nehrkorn et al. (2010). Hourly ERA5 (European ReAnalysis 5) model estimates at 0.25° resolution from the

European Center for Medium-Range Weather Forecasts (ECMWF) are used as input for the WRF model to generate two nested WRF domains. The inner domain covers most of the Upper Rhine Valley with a horizontal resolution of 2 km. The outer domain with a 10 km horizontal resolution includes most of Europe (see Fig. 2, yellow rectangle). STILT is driven by





these nested WRF fields to calculate hourly back-trajectories for 100 released particles with a maximum backward run-time of 72 h for the Heidelberg observation site.


Highly resolved ffCO$_2$ emission inventories from TNO are used to describe the European ffCO$_2$ area and point source emissions separately (Super et al., 2020). The ffCO$_2$ emissions from Germany and its surroundings are resolved on a horizontal grid of about 1 km² (1/60° x 1/120° longitude x latitude). Emissions from the rest of Europe have a horizontal resolution of 0.1° x 0.05°. In the following we explain the mapping of the ffCO$_2$ emissions to the back-trajectories calculated with WRF–
STILT.

### 2.2.1 Surface Source Influence (SSI) approach

According to Lin et al. (2003) concentration changes $\Delta C(\boldsymbol{x}_r, t_r)$ at the observation site at $\boldsymbol{x}_r$ and at time $t_r$ can be described by

$$\Delta C(\boldsymbol{x}_r, t_r) = \int_{t_0}^{t_r} dt \int_V dx\, dy\, dz\, I(\boldsymbol{x}_r, t_r | \boldsymbol{x}, t) \cdot S(\boldsymbol{x}, t), \tag{1}$$

where $S(\boldsymbol{x}, t)$ describes volume ffCO$_2$ sources in [ppm h$^{-1}$] and $I(\boldsymbol{x}_r, t_r | \boldsymbol{x}, t)$ is the influence function for the observation site with units [m$^{-3}$], which links the sources to concentration enhancements. The time and volume integration of the influence function can be realized by tallying the total length of time $\Delta t_{p,m,i,j,k}$ each released particle $p$ spends in a volume element $(i,j,k)$ over time step $m$ (see Lin et al., 2003) and then normalizing to the number of released particles $N_{tot}$:

$$\int_{t_m}^{t_m+\tau} \int_{x_i}^{x_i+\Delta x} dx \int_{y_j}^{y_j+\Delta y} dy \int_{z_k}^{z_k+\Delta z} dz\, I(\boldsymbol{x}_r, t_r | \boldsymbol{x}, t) = \frac{1}{N_{tot}} \sum_{p=1}^{N_{tot}} \Delta t_{p,m,i,j,k}. \tag{2}$$

Moreover, the volume source $S(\boldsymbol{x}, t)$ can be linked to surface fluxes $F(x, y, t)$ in units [mol m$^{-2}$ s$^{-1}$] by assuming that turbulent mixing is strong enough to completely mix the surface emissions from the ground into an air column with height $h$ within one model time step $m$. Commonly, this height $h$ is set to half of the planetary boundary layer height $h_{PBL}$: $h = \frac{1}{2} h_{PBL}$. Then one gets:

$$S(\boldsymbol{x}, t) = \begin{cases} \frac{m_{air}}{h\,\bar{\rho}(x,y,t)}\, F(x, y, t) & \text{for } z \leq h \\ 0 & \text{for } z > h \end{cases}, \tag{3}$$

with the molar mass of air $m_{air}$ and the average air density $\bar{\rho}(x, y, t)$ below $h$. Inserting Eq. (2) and (3) into Eq. (1) yields the contribution from each surface grid cell $(i,j)$ and time step $m$ to the total ffCO$_2$ concentration enhancement $\Delta C(\boldsymbol{x}_r, t_r)$ at the observation site:

$$\Delta C_{m,i,j}(\boldsymbol{x}_r, t_r) = \frac{m_{air}}{h\,\bar{\rho}(x_i,y_j,t_m)} \cdot \frac{1}{N_{tot}} \sum_{p=1}^{N_{tot}} \Delta t_{p,m,i,j,k} \cdot F(x_i, y_j, t_m) \equiv f(\boldsymbol{x}_r, t_r | x_i, y_j, t_m) \cdot F(x_i, y_j, t_m). \tag{4}$$



Here, we call $f(\boldsymbol{x_r}, t_r | x_i, y_j, t_m)$ the footprint or surface source influence element, which connects the surface fluxes from grid

cell $(x_i, y_j)$ at time $t_m$ to a surface source contribution $\Delta C_{m,i,j}(\boldsymbol{x_r}, t_r)$ to the concentration enhancement at the observation site. The sum over all grid cells and times then yields the total concentration enhancement $\Delta C(\boldsymbol{x_r}, t_r)$ at the observation site at $\boldsymbol{x_r}$ and time $t_r$.

### 2.2.2 Volume Source Influence (VSI) approach

Fasoli et al. (2018) showed that nearby area sources in the so-called hyper near field (i.e., within a distance of less than 10 km)

of the observation site are often diluted to only a fraction of the PBLH due to insufficient mixing. Since STILT assumes a complete dilution below $\frac{1}{2} h_{PBL}$ this leads to an underestimation of the contribution of the nearby surface fluxes at the observation site. A solution for this is to calculate an effective mixing depth $h'$ in the hyper near field based on homogeneous turbulence theory (Fasoli et al., 2018; Taylor, 1922).

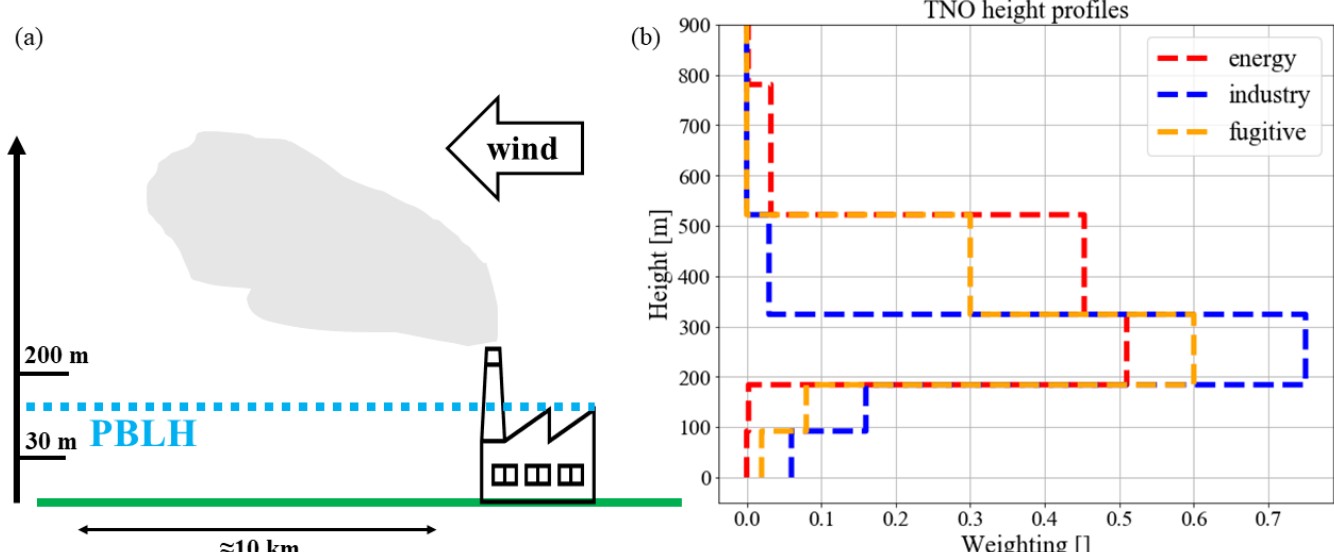

**Figure 3: (a) Sketch of a possible nocturnal situation when the planetary boundary layer height (PBLH) lies above the measurement height at 30 m a.g.l. but below the exhaust of a nearby power plant stack. (b) TNO height profiles for the public power (energy), industry and fugitive sectors, which were used to calculate the volume influences for the associated point sources.**

Here, we focus on nearby point source emissions, which are released from stack heights of up to several hundred meters. Handling these nearby point source emissions as surface fluxes will cause errors in the concentration estimates. Consider e.g. a sample collection at 30 m a.g.l. and a 200 m coal power plant exhaust at a distance of about 10 km, which is the situation at our measurement site in Heidelberg (see sketch in Fig. 3, left panel). During typical summer nights with nocturnal inversions, the emissions of the power plant can be above the planetary boundary layer and its influence on the Heidelberg measurements





would be very small. But in the surface source influence (SSI) approach, where all emissions from this power plant are mixed
into the bottom half of the boundary layer, this will result in large ffCO$_2$ overestimations at the measurement site. To tackle
this problem and improve the representation of nearby point source emissions in STILT, we use sector-specific height profiles
of the point source emissions from TNO and calculate the so-called volume source influence (VSI) for each height interval.
Figure 3 (right panel) shows the discrete TNO emission height profiles for the relevant point source sectors, i.e. those which

are present in the 100 km x 100 km area around Heidelberg. These effective emission heights take the stack heights of the
point sources as well as subsequent plume rise into account (Kuenen et al., 2021).

The point source fluxes $F(x, y, t)$ can be distributed into these individual height intervals $\kappa$ with the TNO sector-specific and
height-dependent weighting factors $g_\kappa$, so that the volume source $S(x, t)$ can be expressed for each height interval $\kappa$ by:

$$S_\kappa(\boldsymbol{x}, t) = V_{mol}(\boldsymbol{x}, t) \cdot \frac{F(x,y,t)}{(z_{\kappa+1}-z_\kappa)} \cdot g_\kappa, \qquad \text{for } z_\kappa \leq z < z_{\kappa+1}. \tag{5}$$

For this, we simply assume the molar volume to be constant throughout the different TNO height intervals (from 0 to 1106
m), i.e., $V_{mol}(x_i, y_j, z_k, t_m) = V_{mol}(x_i, y_j, t_m) = \frac{m_{air}}{\tilde{\rho}(x_i, y_j, t_m)}$, with $\tilde{\rho}(x_i, y_j, t_m)$ being the average of the air densities at the
particle positions in the air column above $(i,j)$ at time step $m$. We now can calculate for each height interval $\kappa$ the contribution
$\Delta C_{\kappa,m,i,j}(x_r, t_r)$ to the total concentration enhancement at the observation site by tallying the total length of time $\Delta t_{p,m,i,j,\kappa}$

each released particle $p$ spends in the volume element $(i,j,\kappa)$ over time step $m$:

$$\Delta C_{\kappa,m,i,j}(x_r, t_r) = \frac{m_{air}}{\tilde{\rho}(x_i, y_j, t_m)} \cdot \frac{1}{N_{tot}} \sum_{p=1}^{N_{tot}} \Delta t_{p,m,i,j,\kappa} \cdot F(x_i, y_j, t_m) \cdot \frac{g_\kappa}{(z_{\kappa+1}-z_\kappa)} \equiv v(x_r, t_r | x_i, y_j, z_\kappa, t_m) \cdot F(x_i, y_j, t_m) \cdot \frac{g_\kappa}{(z_{\kappa+1}-z_\kappa)}. \tag{6}$$

In analogy to the surface source influence, we here call $v(x_r, t_r | x_i, y_j, z_\kappa, t_m)$ the volume source influence and $\Delta C_{\kappa,m,i,j}(x_r, t_r)$
the volume source contribution to the total concentration enhancement at the observation site.


In this study we used the volume source influence approach to model the contributions from the TNO point sources within a
100 km x 100 km box around Heidelberg. All point sources further away as well as the area sources were treated with the
surface source approach.

## 2.3 CO$_2$ sampling for [14]C analysis

Since in Heidelberg separate nighttime (from 18 to 06 UTC) and daytime (from 11 to 16 UTC) two-week integrated CO$_2$
samples for [14]C analysis are available, the model performance can be investigated separately for night and day. The CO$_2$
sampling technique is described in detail by Levin et al. (1980), the analysis technique by Kromer and Münnich (1992). To
estimate regional ffCO$_2$ concentration enhancements from the measured $\Delta^{14}CO_2$, the $\Delta^{14}CO_2$ signature of background air must
be known. Here we use a harmonic fit curve calculated through the $\Delta^{14}CO_2$ observations from Mace Head at the western coast





of Ireland (MHD, 53°20'N, 9°54'W, 25 m a.s.l.) and Izaña on Tenerife Island (IZO 28°18'N, 16°29'W, 2400 m a.s.l.), which
are both presumably mainly influenced by clean Atlantic air masses (at Mace Head only clean Atlantic air masses are collected
for $\Delta^{14}CO_2$ analysis). We assume this marine background to be most comparable to the model ffCO$_2$ background, which is set
to zero at the border of the model domain (Fig. 2, left panel, yellow rectangular). The ffCO$_2$ enhancement $c_{ff}$ based on the
Heidelberg $\Delta^{14}CO_2$ measurements can then be calculated according to

$$c_{ff} = c_{CO_2} \cdot \frac{\Delta^{14}CO_{2,BG} - (\Delta^{14}CO_2 - \Delta^{14}CO_{2,NUC})}{\Delta^{14}CO_{2,BG} + 1000‰},$$   (7)

with $c_{CO_2}$ being the average CO$_2$ concentration in Heidelberg during the two-week integrated sampling period and $\Delta^{14}CO_{2,BG}$
being the $\Delta^{14}CO_2$ signature of background air. The $\Delta^{14}CO_{2,NUC}$ term describes the contributions from $^{14}CO_2$ emissions from
nuclear facilities and is modelled with the volume source influence approach by assuming that all nuclear $^{14}CO_2$ emissions are
released within a 20 m height interval above a typical stack height of 120 m. In order to avoid interference with our results,

we used the VSI approach to calculate the nuclear corrections regardless of whether we later use the VSI or SSI approach for
the comparison between modelled and observed ffCO$_2$. To calculate the nuclear corrections, we used the annual mean $^{14}CO_2$
emissions from the European Commission RAdioactive Discharges Database (RADD, 2021) for the year 2019. We calculated
a mean nuclear contribution of $\Delta^{14}CO_{2,NUC} = 1.3 \pm 0.7$ ‰ and $1.4 \pm 0.7$ ‰ for the daytime and nighttime samples, respectively.
This corresponds to about 7 % of the mean $\Delta^{14}CO_{2,BG} - \Delta^{14}CO_2$ difference between background and measurement site for

both the daytime and nighttime samples. A detailed derivation of equation (7) can be found e.g., in Levin et al. (2003).

## 3 Results

### 3.1 Comparison of observed and modelled ffCO$_2$ in Heidelberg

In the following section we present the ffCO$_2$ concentrations estimated based on the Heidelberg afternoon and nighttime two-
week integrated samples and compare them to two different WRF–STILT model runs, i.e., the surface (SSI) and the volume

source influence (VSI) approach. Figure 4 shows the measured and modelled two-week integrated afternoon (left) and
nighttime (right) ffCO$_2$ enhancements for Heidelberg from July 2018 to June 2020. The black lines show the $\Delta^{14}CO_2$
observation-based ffCO$_2$ concentrations calculated using Eq. 7. They represent the ffCO$_2$ enhancement compared to a maritime
background introduced in Sect. 2.3. During these two years, the two-week integrated regional ffCO$_2$ concentrations of the
afternoon and nighttime samples range from 0.8 to 26.9 ppm and from 2.3 to 23.7 ppm, respectively, with quite similar mean

concentrations of 8.2 ppm in the afternoon and 9.0 ppm during night. Both the afternoon and the nighttime samples show a
clear seasonal cycle, with about three to four times larger ffCO$_2$ concentrations during winter than during summer.

For the afternoon situations, the SSI and the VSI model runs lead to similar root-mean-square deviations (RMSD) between
modelled and measured ffCO$_2$ concentration of 3.7 ppm, considered over the whole two-year period. Whereas the SSI approach





leads on average to a small (10 %) overestimation of the $ffCO_2$ concentrations by 0.8 ppm, the VSI approach tends to underestimate $ffCO_2$ by 0.7 ppm (9 %). The standard deviations of the observation minus model differences are about 4 ppm for both cases. However, there are seasonal differences in the performance of the two approaches. Whereas both model runs lead to a RMSD between modelled and measured $ffCO_2$ concentrations of 2.0 ppm during the summer half year (from April to September), the RMSD during the winter half year (between October and March) is more than twice as high (4.6 ppm and

4.7 ppm in case of the SSI and the VSI approach, respectively). There are, however, differences between the two modelled winters: Whereas the VSI approach leads to an improvement compared to the SSI approach during the winter 2018/2019 (RMSD of 2.9 ppm vs. 4.3 ppm), the subsequent winter 2019/2020 shows poorer performance by both modelling approaches (RMSD of 5.9 ppm for the VSI and RMSD of 4.9 ppm for the SSI approach).




**Figure 4: Comparison of two-week integrated** [14]**C-based measured (black) and modelled (colored) ffCO$_2$ concentration enhancements during afternoon hours (between 11 and 16 UTC; left panels (a) and (b)) and during nighttime (between 18 and 6 UTC; right panels (c) and (d)) for the time period of July 2018 until June 2020 in Heidelberg. Two modelling approaches were tested: the standard surface source influence (SSI) approach (orange; panels (a) and (c)) and the volume source influence (VSI) approach (red; panels (b) and (d)), see text for further details. For each of the comparisons, the root-mean-square deviation (RMSD) between model and observation as well as the mean difference (observation minus model) and the standard error of the mean are given. At the top of each panel the winter and summer periods are marked in blue and green.**

During nighttime situations we observe large differences between the SSI and VSI approaches. The VSI approach leads to a model-data mismatch which is comparable to the afternoon situations, with a mean offset between model and observations of -0.7 ppm (8 %) and a RMSD of 3.4 ppm (the RMSD is 3.3 ppm during summertime and 3.6 ppm during wintertime). In contrast, the nighttime SSI run shows by far the largest ffCO$_2$ overestimations throughout the two years with the largest model-observations deviations during summer (the RMSD is 6.7 ppm during summertime and 5.8 ppm during wintertime). Over the whole two years the average offset is -4.6 ppm (51 %), and the RMSD of 6.3 ppm is almost twice as high as the RMSD of the VSI approach and that of the SSI approach in the afternoon.






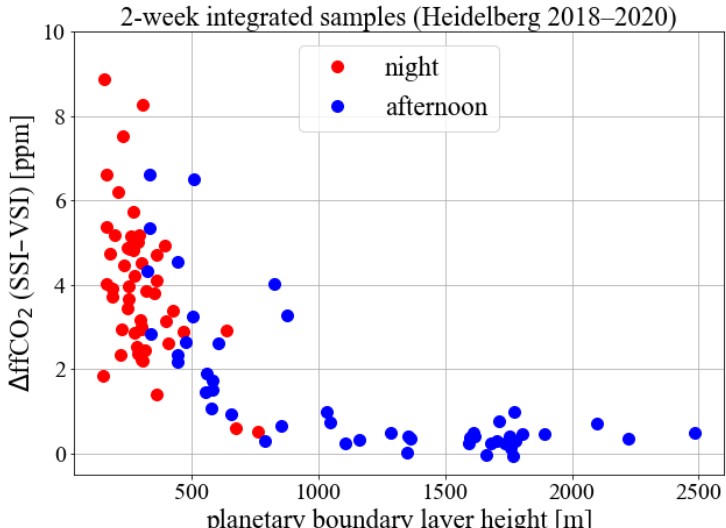

**Figure 5: Modelled ffCO₂ differences between the SSI and VSI approaches for Heidelberg afternoon (blue) and nighttime (red) samples plotted against the modelled mean height of the planetary boundary layer (PBL) during sampling.**


We further investigated why the VSI approach is better than the SSI approach during nighttime, while both approaches are comparable during afternoon situations. For this we extracted the modelled planetary boundary layer height for Heidelberg from the simulations and averaged over the nighttime or afternoon times for the full two weeks. Figure 5 shows the ffCO₂ concentration difference between the SSI and VSI approaches plotted versus the planetary boundary layer height for all two-

week integrated afternoon (in blue) and nighttime (in red) situations over the two years of measurements. During most of the afternoon situations the PBLHs are large, indicating strong convective mixing. The SSI approach with emissions into the bottom half of the PBL then yields similar concentrations at the measurement point as the VSI approach, because the VSI height profiles do not (or only slightly) exceed the bottom half of the PBL. On the other hand, low PBLHs result in large concentration differences between the SSI and VSI approaches, which is the case in most of the nighttime and in some

afternoon situations between mid-October and -February with suppressed convective mixing. During these situations, the SSI approach releases all point source emissions into a shallow layer below the bottom half of the PBL, thus overestimating concentrations at 30 m a.g.l. In contrast, the VSI approach releases emissions at the actual plume height; however due to the shallow PBL and suppressed convective mixing this leads to only small contributions for an observation site inside the PBL (as is the case for low sampling heights such as at the measurement site in Heidelberg).






## 3.2 Surface and volume source contributions from nearby point sources in a "pseudo power plant experiment"

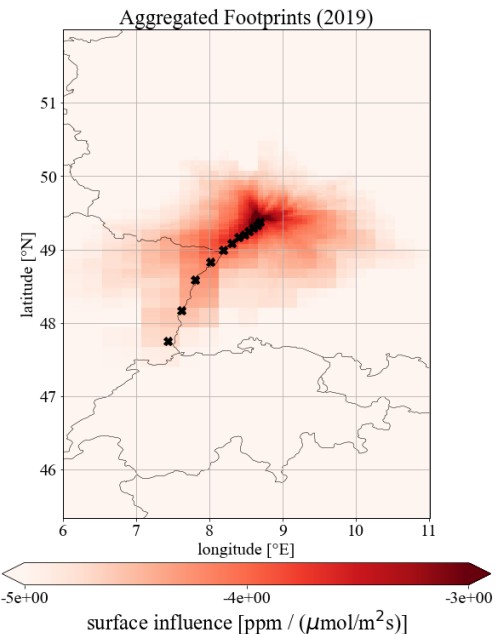

**Figure 6: Aggregated hourly footprints in 2019 for the observation site Heidelberg at 30 m height a.g.l. The black crosses indicate**
**the locations of the 12 pseudo power plants, located at distances of 5, 10, 15, 20, 25, 30, 40, 50, 70, 100, 150 and 200 km from the Heidelberg observation site.**

Next, we wanted to evaluate if the VSI approach is also relevant for typical continental tall tower stations with elevated sampling heights of e.g. 200 m a.g.l. For this we conducted a so-called "pseudo power plant experiment". This experiment
should also help determine up to which distance from the measurement site point source emissions should be modelled with the VSI approach to avoid strong overestimations in modelled concentrations during nighttime. Figure 6 shows the aggregated footprints for Heidelberg in 2019, calculated with our WRF–STILT configuration presented in Sect. 2.2. This mean footprint shows a tail towards the south-western direction, which can be explained by the channeling effect of the Rhine Valley. In our experiment we placed 12 artificial ("pseudo") power plants along this footprint tail at distances of 5 to 200 km from Heidelberg,
as indicated by the black crosses, so that many situations with contributions from these locations reaching the measurement site in Heidelberg could be expected. All power plants were assigned a $CO_2$ emission rate of $10^6$ tons per year, which corresponds to typical emissions of small hard coal power plants in Germany (Fraunhofer, 2021). For every hour in 2019, the ffCO$_2$ contribution from each pseudo power plant was modelled with the SSI and VSI approach. In case of the VSI approach, we used the TNO emission height profile for the public power (energy) sector (see Fig. 3, right panel). We then selected only
those hours for which *each* of the 12 pseudo power plant grid cells were hit by at least one particle back-trajectory. By doing so, we compare the same meteorological situations for the different pseudo power plants and avoid situations when a given



power plant is not in the catchment area of the observation site. This yields 2060 selected hours in 2019. We then extracted

the PBLH at Heidelberg from the WRF–STILT simulation and divided these events into two PBLH regimes (PBLH < 500 m

and PBLH > 500 m). The PBLH < 500 m situations are predominantly nighttime situations, and the PBLH > 500 m are mainly

daytime situations (in 2019 84 % of the nighttime hours have a PBLH < 500 m and 75 % of the daytime situations a PBLH >

500 m).

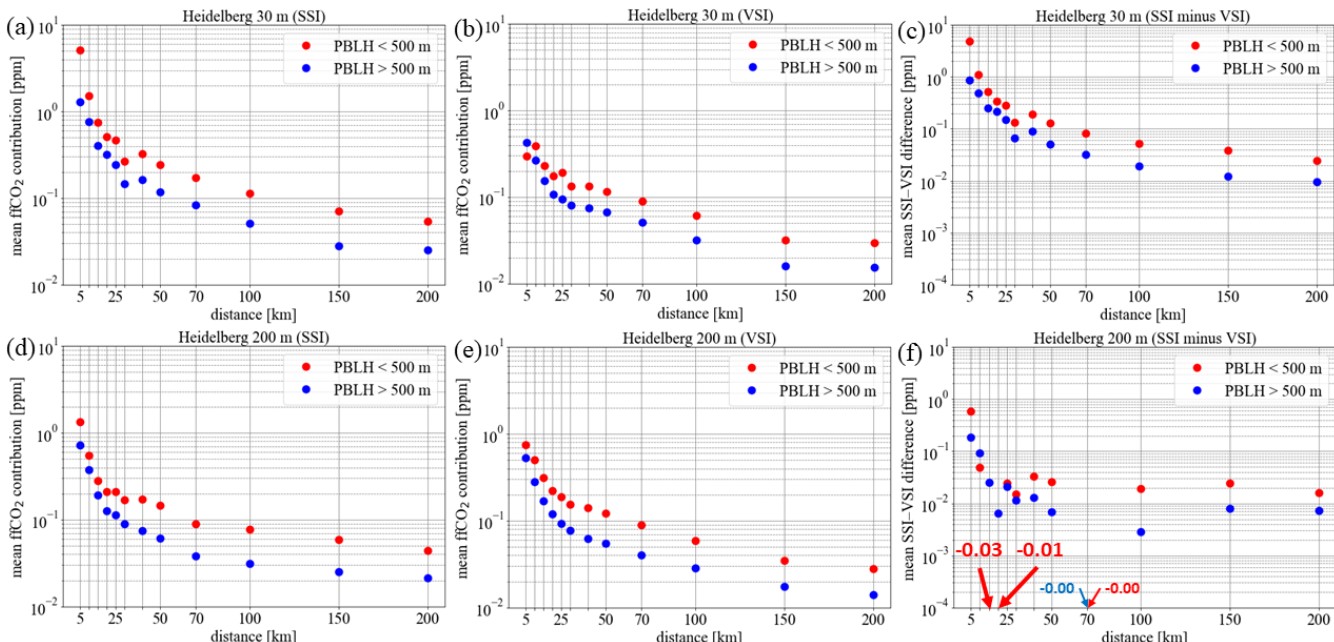

**Figure 7: Mean ffCO₂ contributions from pseudo power plants, which were placed at distances between 5 and 200 km from the**
**observation site Heidelberg at 30 m (upper panels (a) – (c)) and at a virtual 200 m height (lower panels (d) – (f)). Shown are the**
**results from the SSI (left panels (a) and (d)) and VSI approach when using the TNO public power (energy) profile (middle panels**
**(b) and (e)), as well as the mean difference between the SSI and VSI ffCO₂ contributions (right panels (c) and (f)). From all hours in**
**2019, only those situations were selected for which each pseudo power plant grid cell is hit by at least one of the 100 back-trajectories,**
**which were calculated for each hour. These selected hours are then divided into two planetary boundary layer height (PBLH)**
**regimes (blue and red) and averaged. For this, we always used the PBLH at the Heidelberg measurement site at the time when the**
**air parcels from the power plants arrived in Heidelberg. In the lower right panel negative values are indicated with red (PBLH <**
**500 m) and blue (PBLH > 500 m) arrows.**

The upper left (middle) panel of Figure 7 shows the mean ffCO₂ contributions from the individual pseudo power plants versus

their distances from Heidelberg when the SSI (VSI) approach is used. Events were separated into situations when the PBLH

was smaller than 500 m (red dots) or larger than 500 m (blue dots). The mean ffCO₂ contribution differences between the SSI

and VSI approach (SSI minus VSI) for the individual pseudo power plants are shown in the upper right panel. It is obvious

that the mean ffCO₂ contributions from the power plants decrease with increasing distance from the observation site in both

modelling approaches. This can be explained by the dispersion of the power plant plumes and the associated dilution. To





restrict the mean ffCO$_2$ contribution from these power plants to below 0.1 ppm, the observation site should be more than 100

km (SSI) or 50 km (VSI) away from this power plant. This is in line with the ICOS recommendations that suggest a distance

of at least 40 km from strong anthropogenic sources (ICOS RI, 2020). The upper left panel in Fig. 7 shows that the SSI

approach yields larger contributions for stable PBLH < 500 m situations compared to (daytime) situations with PBLH > 500

m. Since in the SSI approach the emissions are homogeneously mixed into the bottom half of the PBL, the smaller mixing

volume during PBLH < 500 m situations leads to larger ffCO$_2$ concentrations. This is what we already have seen from our

daytime and nighttime simulations of real-world ffCO$_2$ (see Fig. 5). The reduction of the ffCO$_2$ contributions with increasing

PBLH could be seen as an increased vertical dispersion of the power plant plumes. The VSI approach shows the same behavior

with larger ffCO$_2$ contributions during stable PBLH < 500 m situations for most power plants, which can also be explained by

less dispersion of the power plant plumes. However, the power plant within a 5 km radius yields lower ffCO$_2$ contributions

during stable PBLH < 500 m conditions than during PBLH > 500 m situations. A possible explanation is that during stable

PBL conditions the mixing is too weak to transport the emissions from the power plant stack down to the sampling height at

30 m within the time the air mass needs to travel the 5 km from the power plant to the observation site (see Fasoli et al., 2018).

Looking at the mean ffCO$_2$ contribution *differences* (upper right panel of Fig. 7) between the two model approaches reveals

355   that the SSI approach simulates on average almost 5 ppm larger ffCO$_2$ contributions than the VSI approach for the closest

power plant during stable conditions. During PBLH > 500 m situations and for more distant power plants the mean difference

between the SSI and VSI contributions decreases due to stronger mixing or more time for mixing over the longer air mass

travel time between the power plant and observation site. In both cases, the assumption in the SSI approach, i.e. an

instantaneous and homogeneous dilution of all power plant emissions in the bottom half of the PBL seems to be more justified

360   than during PBLH < 500 m situations and for power plants very close to the measurement site.

Since ICOS tower stations have most of their air inlets above 30 m a.g.l., we also investigated the behavior of the SSI and VSI

approach for a virtual Heidelberg sampling height at 200 m a.g.l. The results are shown in the lower panels of Fig. 7. The SSI

approach shows less enhancements compared to the VSI approach during stable conditions and for power plants very close

365   by. This could be explained by situations with very stable conditions (with for PBLH < 200 m), when the sampling height at

200 m a.g.l. is above the PBL and hardly sensitive to emissions, which are mixed within the bottom half of the PBL. In contrast,

the VSI approach yields larger ffCO$_2$ contributions from nearby power plants compared to the case with the 30 m sampling

height, since the sampling height (200 m a.g.l.) is now closer to the effective emission height. Consequently, the 200 m

sampling height shows on average less ffCO$_2$ contribution differences between SSI and VSI approach, especially for

contributions from very close power plants and during stable PBL situations.





## 4 Discussion

### 4.1 Effects of emission uncertainties on the comparison between observed and modelled ffCO$_2$ in Heidelberg

The model-data mismatch presented in Fig. 4 depends not only on the representation of atmospheric transport and the handling of point source emissions, but also on uncertainties in the emission inventory. Since we interpret the model-data mismatch

difference for the evaluation of the SSI and VSI approach, we need to ensure that it is not caused by incorrect area/point source distribution or temporal profiles in the emission inventory. If, for example, the nocturnal point source emissions were overestimated in the inventory, we would, by mistake, consider the VSI approach to yield better agreement with observations for the wrong reason. Therefore, we first want to discuss uncertainties in the inventory, and assess which theoretical overestimation in the inventory would be needed to generate the apparent improvement of the model-data mismatch going

from SSI to the VSI approach. Super et al. (2020) identified four sources of uncertainties in the high-resolution TNO inventory: (1) uncertainties in the national activity data, (2) uncertainties in the emission factors, which quantify the ffCO$_2$ emissions that are released per unit of activity and are related to the carbon content of the fuels, (3) uncertainties in the spatial distribution of the national emissions, which rely on spatial proxies like population or traffic density and, finally, (4) uncertainties in the temporal profiles of emissions. Super et al. (2020) used a Monte-Carlo approach to produce 10 high-resolution TNO inventory

ensembles for the annual emissions in 2015 by incorporating the uncertainties (1) to (3) for the area sources. They regard the point source emission uncertainties as quite low and thus excluded them from the Monte-Carlo simulations. For a 100 km x 100 km area around Heidelberg, the annual total ffCO$_2$ area source emission calculated from the 10 emission grid realizations spreads by about $\pm3$ %. Based on the results of Super et al. (2020) we may thus assume a very low uncertainty for the area and point sources, which could not explain the observed differences in the model-data mismatch between SSI and VSI.


In a thought experiment we tested how much we would have to change the actual point source emissions so that SSI and VSI approach lead to a similarly good agreement with observations during nighttime. In Fig. 8 we show that the point source emissions would have to be reduced by as much as 70 % during nighttime to show a similar model-data mismatch for the SSI approach as for the VSI approach. Such large point source emission uncertainties are unrealistic and unexpected. Based on

these considerations, we conclude that it is highly unlikely that the improved model-data mismatch of the nocturnal VSI approach is due to biases in the temporal profile of the emissions. The improvement in the VSI approach can therefore be attributed to the different vertical representation of the point sources.

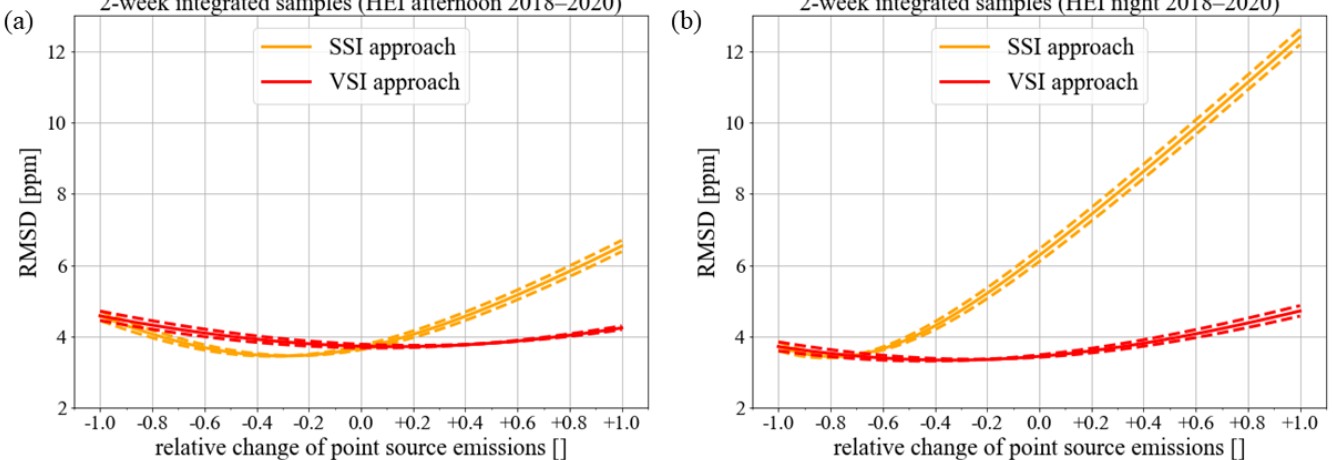

**Figure 8: Root-mean-square deviation (RMSD) between measured and modelled ffCO$_2$ concentration of two-week integrated afternoon (left panel (a)) and nighttime (right panel (b)) $\Delta^{14}$CO$_2$ samples collected during July 2018 and June 2020 in Heidelberg (HEI) at 30 m a.g.l. for the surface (SSI, in orange) and volume source influence (VSI, in red) approach for different relative changes in the TNO point source emissions. A relative change of -1 means that all point source emissions are switched off, and a relative change of +1 means that the actual emissions of all point sources are doubled. For instance, the actual point source emissions would have to be decreased by about 70 % (corresponds to -0.7 on the x-axis), so that SSI and VSI approach lead to a similar RMSD for nighttime situations. The dashed lines show the additional impact of a TNO area source emission uncertainty of ±3 % (see Super et al., 2020) on the RMSD between measured and modelled ffCO$_2$ concentration.**

## 4.2 Applications of the volume source influence approach

Typically, flask samples for model-observation comparisons or inversions are collected in the afternoon during well mixed conditions when the atmospheric transport and mixing processes can be simulated best (Geels et al., 2007). However, the inclusion of nighttime observations into inversion modelling frameworks would drastically increase the number of observational data that could be used to optimize emissions and could help draw conclusions about the mixture and the diurnal emission profiles of source sectors that are more active during night or in the morning and evening hours. Here, we discuss the effect of point source emission heights on the model-data mismatch, especially during nighttime, and assess when and where the volume source influence approach should be applied.

The pseudo power plant experiment yields a mean SSI minus VSI contribution difference between about 0.5 ppm (for a 15 km distant power plant) and 4.9 ppm (for a 5 km distant power plant) during stable conditions with low PBLHs. Since the Heidelberg measurement site is surrounded by several point sources, some of them emitting more than $10^6$ tons CO$_2$ per year (see Fig. 2), we decided to apply the VSI approach to all point sources within a 100 km x 100 km area around Heidelberg and use the SSI approach for the point sources further away, where we expect only small differences between the VSI and SSI approach. The ffCO$_2$ results for the two-week integrated nighttime samples showed that the model-data mismatch could already



be reduced by about 3 ppm (RMSD = 3.4 ppm) when using this VSI approach for nearby point sources instead of the standard SSI approach (RMSD = 6.3 ppm). During well-mixed conditions the pseudo power plant experiment showed less differences

between the VSI and SSI approach, which can also be seen in the $ffCO_2$ results for the two-week integrated afternoon samples, where the VSI approach and the SSI approach differ by merely ca. 1 % (both approaches lead to a RMSD of about 3.7 ppm). Thus, we strongly recommend the application of the VSI approach for measurement sites with sampling heights typically within the nocturnal boundary layer and with nearby point sources so that also nighttime observations could be used, e.g. for a model-observation comparison. However, the VSI approach is accompanied by larger computational costs since the volume

influence field $v$ must be calculated for each height interval. In contrast, in the SSI approach only one surface influence field $f$ must be calculated (see Sect. 2.2). To save computational power we therefore suggest that the VSI approach only be used for nearby point sources and to use the SSI approach for more distant point sources where both model approaches lead to similar results. Depending on the distribution and the emission strength of the point sources around the measurement site and the intake-height of the measurement site, the results from the pseudo power plant experiment can help to decide for which

point sources the VSI approach should be applied. From this experiment it follows that for low intake-heights (e.g. 30 m) and power plants within a radius of 5 to 15 km the SSI minus VSI differences are substantial. When averaged over the two PBLH regimes (< 500 m and > 500 m), these differences come to 3.9 and 0.5 ppm respectively, equivalent to a 12- or 2-fold increase in the absolute VSI contribution for a point source emitting 1 $MtCO_2$ per year. Such a station and point source configuration is realistic for urban observations. For ICOS-like background stations, which should typically be located 50 km from point

sources, the SSI minus VSI difference is less than 0.1 ppm and thus even less than the World Meteorological Organization (WMO) compatibility goal for $CO_2$ (WMO, 2018).

Since the $^{14}CO_2$ samples are collected at many ICOS stations from a higher intake, we performed the pseudo power plant experiment also for a (virtual) Heidelberg observation site at 200 m a.g.l. (where we do not have real measurements). The

results show that for nearby power plants the mean SSI minus VSI contribution differences are roughly one order of magnitude smaller than in the case of the observation site at 30 m a.g.l. However, one has to keep in mind that, although the SSI minus VSI contribution differences are smaller in the case of the 200 m high observation site, the SSI approach does not represent the atmospheric transport processes any better than in the case of the observation site at 30 m a.g.l. It simply means that the 200 m intake height is less sensitive to the bottom half of the PBL during stable conditions, which leads to less overestimations

for the SSI compared to the VSI approach. The randomness of the SSI contributions becomes immediately clear if one considers the 15 km and 20 km distant power plant. Here, the SSI approach yields even smaller contributions than the VSI approach during stable conditions. Moreover, the 200 m intake height is vertically closer to the effective emission height of the power plants, which leads to larger VSI contributions compared to the 30 m level. These two circumstances cause the smaller mean SSI minus VSI contribution differences for nearby point sources in the case of the 200 m level. The mean SSI

minus VSI contribution difference for a $10^6$ tons $CO_2$ per year emitting point source is below 0.1 ppm if the point source is at least 10 km away from the measurement site. However, one has to keep in mind that this absolute difference in SSI minus VSI



contribution increases linearly with the emission strength of the point sources. Thus, for ICOS-like stations and point sources at least 10 km away, the SSI approach again seems to be well suited when there is enough time for mixing throughout the PBL and the SSI assumptions are justified.


Inaccurately representation of point source emissions from stacks is not limited to Lagrangian models, but is found in many Eulerian modelling setups as well. Super et al. (2017) investigated how well a Eulerian model (WRF–Chem) alone as well as in combination with a Gaussian plume model agrees to $CO_2$ and CO mixing ratios at an urban site in the Netherlands. In the case of the Eulerian model the point source emissions are distributed over the different vertical model levels according to the

emission height profiles shown in Fig. 3, which is rather similar to the VSI approach we used in WRF–STILT. The Gaussian plume model is able to represent the exact emission stack heights and improves the description of the transport and dispersion of the point source plumes, which are in the case of Eulerian models instantly mixed within individual grid boxes (Super et al., 2017). The authors could show that both the exact representation of the stack heights as well as the more appropriate description of the plume dispersion will lead to a better agreement to the observations in the case of the WRF–Chem model in

combination with the Gaussian plume model. Therefore, they recommend to treat all large point source emissions within a 10 km radius around the observation site with such a plume model.

## 5 Conclusions

In this study we used a two-year record of afternoon and nighttime two-week integrated [14]C-based $ffCO_2$ measurements

conducted in Heidelberg, at 30 m a.g.l., to examine the performance of the standard STILT surface source influence (SSI) approach. We find that it is almost twice as good for afternoon situations (RMSD = 3.7 ppm) than for the nighttime situations (RMSD = 6.3 ppm) when comparing modelled and observed $ffCO_2$ concentrations. The lower performance during night could be explained by the large overestimation of the contributions from nearby point sources. We therefore introduced an alternative modelling approach – the volume source influence (VSI) approach – which is able to represent the emission height and the

plume rise of the point source emissions more correctly. With this approach, the performance of STILT is similar for the afternoon (RMSD = 3.7 ppm) and nighttime samples (RMSD = 3.4 ppm).

We further investigated the behavior of the SSI and VSI approach for point sources at different distances to the measurement site and under different atmospheric conditions. For this we performed a pseudo power plant experiment by modelling the

$ffCO_2$ contributions from 12 virtual power plants, each emitting one million tons of $CO_2$ per year and placed 5 to 200 km away from the observation site. This model experiment could confirm what we already observed in the model-observation comparison of the two-week integrated samples, namely that the standard SSI approach leads to strong overestimations compared to the VSI approach given stable atmospheric conditions with low planetary boundary layer heights, especially for



point sources close to the observation site. For instance, point sources with a distance between 5 and 15 km to the observation

site lead to a mean SSI minus VSI difference of 3.9 to 0.5 ppm ffCO$_2$, which is 12 to 2 times larger than the mean VSI ffCO$_2$ contribution from these point sources. Thus, we strongly recommend the use of the VSI approach for these close-by point sources when modelling their ffCO$_2$ contribution at low altitude measurement sites. For ICOS-like background stations, which should typically be located more than 50 km away from point sources, the mean SSI minus VSI difference reduces to below 0.1 ppm. We also performed this model experiment for a virtual observation site with a 200 m sampling height, which is more

comparable to the uppermost measurement height of typical ICOS stations. Here, the mean contribution differences between the SSI and VSI approaches for nearby point sources are smaller compared to that at the 30 m sampling height, because the 200 m height is less sensitive to the bottom half of the PBL during very stable situations (leading to smaller SSI contributions) and is vertically closer to the effective power plant emission height (leading to larger VSI contributions). Whereas for low sampling heights the VSI approach is strongly recommended to model contributions from nearby point sources in order to

avoid large overestimations (in the order of several ppm for ffCO$_2$) during stable conditions, we also suggest the use of the VSI approach in the case of sampling heights well above the nocturnal boundary layer since it is the physically more correct approach for these situations with suppressed mixing. The contributions from more distant point sources are generally smaller and also the assumptions in the SSI approach seem to be more justified for longer air mass travel times between the point source and observation site and during unstable atmospheric conditions. This explains the smaller differences between the SSI

and VSI approach for these situations. Depending on the atmospheric conditions, the sampling height, the distance to the point source as well as the emission strength of the point source, the results of our pseudo power plant experiment can be used to assess the contribution of the point source in both modelling approaches. Then one can decide if the SSI approach is sufficient (e.g. for distant point sources with lower emissions or during unstable conditions) or if the VSI approach is the better alternative.


Whereas the modelling of transport and mixing processes is still challenging during nighttime, we showed with this study that using the VSI approach for nearby point sources will greatly reduce the overestimations of contributions from nearby point source emissions during periods with low PBLH, especially for low altitude measurement sites. Therefore, this approach could possibly be a first step towards the usage of nighttime observations for modelling purposes in STILT.

**Code and data availability**

The measurement and model results for the two-week integrated samples collected at Heidelberg as well as the outcome of the pseudo power plant experiment are available at the Heidelberg University data depository (https://doi.org/10.11588/data/CK3ZTX). The R script ("volume.infl.ffco2.timeres.r") to calculate ffCO$_2$ contributions with the volume source influence approach has been added to the STILT repository (revision number 747). The STILT model can

be downloaded at http://stilt-model.org/ after registration. The used input fields for STILT are large-sized objects (>2.5 GB

per day) and stored on the Mistral server from the "Deutsches Klimarechenzentrum" (DKRZ, https://www.dkrz.de/up/systems/mistral). These data will be made available upon request.

**Author contribution**

FM designed the study together with CG, IL and SH. FM performed the STILT modelling and evaluated the data. CG helped
with the implementation of STILT. SH compiled the measurement results of the two-week integrated $^{14}CO_2$ samples. IS was responsible for the TNO emission inventories. JM generated the highly resolved meteorological fields with WRF. FM wrote the manuscript with help of all co-authors.

**Competing interests**

The authors declare that they have no conflict of interest.

**Acknowledgement**

The authors gratefully acknowledge Thomas Koch and Michał Gałkowski for their help with running STILT. We wish to thank the staff of TNO at the Department of Climate, Air and Sustainability in Utrecht for the emission inventories and height profiles. A special thank goes to Sabine Kühr and the whole staff of the ICOS-CRL Karl Otto Münnich Laboratory for their careful $^{14}CO_2$ sampling and analysis and to Julian Della Coletta and the ICOS Atmospheric Thematic Centre for conducting
and evaluating the continuous $CO_2$ measurements in Heidelberg. We further would like to thank Ida Storm and the members of the ICOS Carbon Portal for their cooperation in developing tools for estimating nuclear $^{14}CO_2$ contaminations at European ICOS stations. The ICOS Central Radiocarbon Laboratory is funded by the German Federal Ministry of Transport and Digital Infrastructure. FM was paid by the German Weather Service (DWD).

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
