# Peer review of "Effects of point source emission heights in WRF-STILT: a step towards exploiting nocturnal observations in models"

_Geoscientific Model Development, 2021_

## Author Response (AR1)

**Reply to comments of Reviewer 1, Sharon Gourdji**

We want to thank Sharon Gourdji for the review of our manuscript and her helpful and inspiring comments and suggestions for improvement. Our replies are marked in blue.

This is an excellent study, which is very well-written and clear and with very useful implications for atmospheric inverse modeling, particularly in urban areas. A few small questions and concerns for clarification should be addressed before final publication:

 To use the VSI approach, does one also need an inventory containing the vertical height profiles of all point source emissions? This would be great to have, but in practice, this may currently exist in Europe only. (For example, I don't believe that the Vulcan product for the USA contains height of emissions sources now, nor other products like FFDAS or ODIAC.)

Yes, that is correct. The VSI approach is based on vertical height profiles of the point source emissions. Ideally, one would have an inventory for the effective emission heights of all individual point sources (in the surroundings of the measurement site), which consider the actual stack heights plus plume rise. In our study we have used source sector-specific (average) emission height profiles from TNO, which are representative for Europe. While checking the Vulcan, FFDAS or ODIAC inventories, we didn't find similar emission height profiles for the USA so far. However, applying the TNO height profiles also in the USA could already lead to a first improvement compared to releasing all point source emissions from ground. But we agree that such an inventory containing the vertical point source emission heights for the whole globe would be important to have. We have added a sentence to our conclusions in the revised manuscript (lines 583ff).

• What are the additional computational requirements of the VSI relative to the SSI approach? Also, how would one go about creating a footprint from a single tower with a mix of the VSI approach for nearby point sources and the SSI for farther-away emissions sources? How would one do that practically with the WRF-STILT framework?

In STILT, the trajectory information of the released particles is saved in RData files. In the SSI approach the number of particles below the half of the boundary layer height is counted in each grid cell and is then used to weight the surface fluxes in the respective grid cell. In the VSI approach we used vertical emission height profiles with seven different height intervals (see Fig. 3b in our manuscript). This means that in the VSI approach the number of particles in seven (instead of one) height intervals must be counted, which results in additional computational costs. That's why we recommend using the VSI approach only for nearby point sources (where it matters). For this, one could first set the nearby point sources to zero and calculate the contributions from the area sources and the farther-away point sources with the standard SSI approach. Then, one could use our R script provided in https://doi.org/10.5281/zenodo.5911518 to calculate separately the volume source influences for the seven height intervals in the near field of the station to get the contributions from the nearby point sources.

• I was left wondering what are the relative impacts of mixing assumptions versus PBL height errors when using night-time measurements. Could you include a small theoretical example to demonstrate the impact of realistic mixing height errors with the VSI approach and nighttime observations?

The SSI approach assumes that the air masses within the bottom half of the PBL height ( $h_{PBLH}$ ) are well mixed. Thus, the surface emissions are weighted with  $1/(\frac{1}{2}h_{PBLH})$ . If the nocturnal PBL heights, and so the strength of the mixing, are overestimated (as it is the case for the ECMWF-derived mixing heights in Gerbig et al., 2008) this would result in an underestimation of the contribution from those surface emissions. For daytime situations, Gerbig et al. (2008) showed the propagated impact of those mixing height uncertainties on the modelled CO2 mixing ratios by introducing (besides the turbulent winds) a second stochastic process, which rescales the footprint (i.e. the sensitivity to surface fluxes) and thus considers the mixing height uncertainty. Their ansatz would be more difficult for nighttime situations, which have much larger mixing heights biases.

The VSI approach assumes well-mixed conditions in each of the seven fixed height intervals of the TNO vertical emission height profiles. But it also depends on the mixing to be correct. Imagine a nighttime situation with a too shallow modelled PBL height. This would imply that mixing is insufficient, and the tracer increments are overestimated within the PBL. If a power plant plume is within the PBL, also the VSI approach will yield too large power plant CO2 contributions. Thus, also the VSI approach suffers from an incorrect representation of the PBL height.

**Other small comments:**

• Abstract, line 28: "to fall below 0.1 ppm" à during day or nighttime or both?

Thank you for this hint! This corresponds to situations with PBL heights smaller than 500 m (see Fig. 7c in the manuscript). We have specified it in the revised manuscript (line 30).

• Page 3, line 61: "nighttime situations showed a relative bias of more than 50%" -> in which direction is this bias?

This positive bias means that the ECMWF-derived mixing heights are larger than the mixing heights estimated from radiosonde data. We have clarified this in the revised manuscript (lines 64f).

• Is 100 particles enough for this study? I assume you would get the same results using 500 particles or more, but it might be worth a small check for sensitivity here.

Thank your for pointing us to this sensitivity study. To check this, we calculated the ffCO2 contributions from the 12 pseudo power plants for the 30 m high receptor site Heidelberg by having released 500 particles. To save computational power, we considered only one month (January) in 2019. Fig. 1 shows the mean VSI minus SSI contribution differences (as Fig. 7c in the manuscript) for January 2019. The differences between 100 and 500 released particles are usually only very small. So, we can argue that 500 released particles would not change the overall picture and 100 particles are enough.

Figure 1: Mean SSI minus VSI difference in  $ffCO_2$  contributions from pseudo power plants, which were placed at distances between 5 and 200 km from the observation site Heidelberg at 30 m. The time period is January 2019. Panel (a) shows the results if 100 particles are released each hour and panel (b) shows the results for 500 released particles per hour. For further details, please refer to the caption of Fig. 7 in the manuscript.

• Figure 1: This is a nice map, although it's a bit hard to see the country outlines and the actual distance from point sources to measurement locations. Consider additionally including a histogram or barplot of distance to nearest point source(s) for each measurement location? To what extent do existing measurement locations follow the ICOS recommendations to stay at least 40 km away from strong anthropogenic sources? (And how did ICOS derive this recommendation in the first place?)

The ICOS recommendation (https://doi.org/10.18160/GK28-2188) of 40 km distance between the station and strong anthropogenic sources was chosen to "ensure that observations can be represented in atmospheric transport models with spatial resolution of around 10-20 km". As can be seen in Fig. 1 in our manuscript, not all ICOS stations fulfill this recommendation. In those

cases, "a footprint and representativeness study should be performed". We have included a table, which sums up the point source emissions in a 50 km x 50 km box around the most affected ICOS stations in Fig. 1 in our revised manuscript.

• Page 3, line 61: "a relative standard deviation of about 40%" in mixing height, or errors in mixing height? Also, please clarify for following sentence.

This means the relative standard deviation of the difference between the ECMWF-derived mixing heights  $z_i(ECMWF)$  and the radiosonde estimates  $z_i(RS)$ , i.e.  $\frac{std(z_i(ECMWF)-z_i(RS))}{\langle z_i(RS) \rangle}$ . We have clarified this in the revised manuscript (line 63).

 Page 4, lines 62-64: if the uncertainty in daytime mixing height translates into uncertainties of ~3 ppm and 30% of the simulated biogenic signal during summer, what does this tell you about nighttime uncertainties? Just complete the thought here. Also, in reference to the previous comment, this article develops a better approach to dealing with mixing assumptions in STILT but doesn't address or improve mixing height errors. So, what is the relative impact of these two types of errors on both daytime and nighttime measurements?

The authors of this cited study (Gerbig et al., 2008) only investigated the propagation of uncertainties in mixing heights into mixing ratios during daytime situations. They state that nighttime mixing heights have much larger uncertainties and biases, which makes them much more difficult to consider in STILT. Nevertheless, we would expect much larger uncertainties in the mixing ratios during nighttime.

We want to focus in our study on the difference between the mixing assumptions in the SSI and VSI approach and the improvements, which can be achieved when just using the optimized mixing assumptions of the VSI approach instead of the standard SSI approach. These improvements of the VSI approach can be seen for example in Fig. 4 and 7 in our manuscript, which also distinguish between daytime and nighttime situations.

• Page 5, lines 88-95: this is a great explanation for why the ability to use nighttime observations in inversions would be very useful and is a prime rationale for your study. I suggest adding a statement to this effect in the abstract about why this work would be very helpful for other researchers for the reasons laid out here.

Thank you for this suggestion! We have added a sentence in the abstract (lines 24f).

• Page 7, line 141: please spell out what TNO stands for, for those not familiar. In general, it might be nice to describe this inventory in a bit more detail for non-European audiences, especially because you are relying on the height profiles in this inventory to implement your VSI approach. Also, for the differing spatial resolutions between Germany and the rest of Europe, is this how it's produced in Super et al, 2020, or do you aggregate emissions yourself for the purposes of this study?

TNO stands for the Netherlands Organisation for Applied Scientific Research. We have added a bit more information about the TNO inventories in the revised manuscript (lines 164ff). There are two inventories with different horizontal resolutions available: one highresolution inventory (1/60° x 1/120°) for Germany and its surroundings and one low-resolution inventory (0.1° x 0.05°) for most of Europe. In our study, we have just nested these two inventories.

• Page 9, lines 189-191: How would time-varying emissions affect these TNO height profiles (e.g. with some emission sources starting and stopping again)? Also, do the TNO height profiles shown in Figure 3b represent sector-specific averages? Or are heights included for individual point source locations as well?

We considered in our study time-varying point source emissions. For this we used the source sector-specific diurnal, weekly and seasonal temporal profiles from TNO (see Fig. 2 below). Yes, the height profiles shown in Fig. 3b in the manuscript represent sector-specific averages, which should be representative for Europe. Unfortunately, there are no stack heights for individual point source locations available. We have added a few sentences in our revised manuscript (lines 226ff).